# On the Edge of Dispensability, the Chloroplast *ndh* Genes

**DOI:** 10.3390/ijms222212505

**Published:** 2021-11-19

**Authors:** Bartolomé Sabater

**Affiliations:** Department of Life Sciences, University of Alcalá, Alcalá de Henares, 28805 Madrid, Spain; bartolome.sabater@hotmail.com; Tel.: +34-609-227-010

**Keywords:** Ndh complex, photosynthesis, plant evolution, plastid DNA, stress protection

## Abstract

The polypeptides encoded by the chloroplast *ndh* genes and some nuclear genes form the thylakoid NADH dehydrogenase (Ndh) complex, homologous to the mitochondrial complex I. Except for Charophyceae (algae related to higher plants) and a few Prasinophyceae, all eukaryotic algae lack *ndh* genes. Among vascular plants, the *ndh* genes are absent in epiphytic and in some species scattered among different genera, families, and orders. The recent identification of many plants lacking plastid *ndh* genes allows comparison on phylogenetic trees and functional investigations of the *ndh* genes. The *ndh* genes protect Angiosperms under various terrestrial stresses, maintaining efficient photosynthesis. On the edge of dispensability, *ndh* genes provide a test for the natural selection of photosynthesis-related genes in evolution. Variable evolutionary environments place Angiosperms without *ndh* genes at risk of extinction and, probably, most extant ones may have lost *ndh* genes recently. Therefore, they are evolutionary endpoints in phylogenetic trees. The low number of sequenced plastid DNA and the long lifespan of some Gymnosperms lacking *ndh* genes challenge models about the role of *ndh* genes protecting against stress and promoting leaf senescence. Additional DNA sequencing in Gymnosperms and investigations into the molecular mechanisms of their response to stress will provide a unified model of the evolutionary and functional consequences of the lack of *ndh* genes.

## 1. Introduction

The *ndh* genes are homologous to those encoding components of mitochondrial and bacterial respiratory complex I (NADH dehydrogenase, EC 1.6.5.3). Their identification in chloroplast DNA, by the 1980s [1,2], was a surprise because the respiratory electron transport chain and the photosynthetic electron transport chain are characteristic of mitochondria and chloroplast respectively, and there was no evidence for the presence of any complex I-like or respiratory-like process in chloroplasts. Therefore, the role, if any, of the *ndh* genes in the chloroplast became an active field of research that continues today.

Chloroplasts evolutionarily derive from primitive endosymbiont cyanobacteria in host cells [3,4]. Many genes from cyanobacteria were progressively transferred to the nucleus of host cells, and the engulfed cyanobacteria evolved into chloroplasts that are only partially autonomous. Most chloroplast proteins are encoded in nuclear DNA, and only a few chloroplast proteins (about a hundred) are encoded in genes retained in plastid DNA; among them, the *ndh* genes. These genes, although lost in most algae lineages, are conserved in the plastid DNAs in the phylum Streptophyta and in the derived land plants. This suggests that *ndh* genes provide some advantages for the adaptation from aquatic to terrestrial environments. Figure 1 compares the chloroplast DNAs of *Arabidopsis thaliana* (154.5 kbp), as a model Angiosperm, and *Synechocystis* (3600 kbp), as a model cyanobacterium. Transpositions of the *ndh* genes (yellow) accompanied the reduction of about 95% of the cyanobacterial genome size to become the chloroplast DNA of angiosperms. The *ndh* genes are present in the plastid DNA of bryophytes, ferns, and photosynthetic higher plants, except for a few species that will be considered here. The conservation and the expression of *ndh* genes in chloroplasts vary among photosynthetic eukaryotes, revealing how environmental factors determine the conservation or the loss of genes through natural selection.

Indeed, sequencing of chloroplast DNA from many plants, and immunological and proteomic identification of chloroplast proteins and protein complexes, have demonstrated the presence of *ndh* genes and a complex I-like (the Ndh complex) in the chloroplast of most land plants. Parasitic angiosperms, such as *Epifagus virginiana* and *Cuscuta reflexa*, which have low or no photosynthetic activity, were soon found to lack *ndh* genes [5,6,7], which suggested that the protein products of the *ndh* genes play a role in photosynthesis. However, as we will see below, some fully photosynthetic plants have been found to lack the plastid *ndh* genes and the Ndh complex, making their removal as intriguing as their functional role. The low relative amount of Ndh protein (about 0.2% of the thylakoid protein) [8,9] and the absence of chloroplast *ndh* genes in several plants [10] suggest that the functional role of the Ndh complex in the chloroplast might be dispensable in some environments and in some plant lines.

Early evidence [11,12] still suggested that *ndh* genes are involved in the response to different stresses, which could be related to the evolutionary loss of *ndh* genes. However, other functions of the Ndh complex in photosynthesis and leaf senescence could explain the loss of *ndh* genes. Plastid DNA from many plants have recently been sequenced and provide extensive taxonomic and phylogenetic information on the loss of plastid *ndh* genes. This review will seek correlations of biochemical, functional, and protective properties attributable to the Ndh complex with the estimated time and mode of evolutionary loss of *ndh* genes.

## 2. Chloroplast *ndh* Genes

Table 1 shows the usual name of the 11 chloroplast *ndh* genes (*ndhA* to *ndhK*), their encoded polypeptides, and their homologous subunits in respiratory complex I of mitochondria and respiratory bacteria. In most plants, the *ndh H*, *A*, *I*, *G*, *E*, and *D* genes with the *psaC* gene (encoding the photosystem I protein PsaC and located between *ndhE* and *ndhD* genes) are organized into a transcriptional unit (operon) that maps into the small single-copy region of plastid DNA. The *ndhC*, *K*, and *J* genes constitute another operon located in the large single-copy region of plastid DNA (Figure 1). The *ndhF* (mapping into the small single-copy region) and two identical copies of the *ndhB* gene (one in each of the two inverted repeated regions) are transcribed independently as monocistronic mRNA. Transpositions, inversions, and deletions within the DNA strand account for the deviations from the canonical gene organization, prevalent in Angiosperms (Figure 1). Each of the *ndhA* and *ndhB* genes has an intron of approximately 700 and 1000 nucleotides, respectively.

Consistent with a functional role of the proteins encoded by the *ndh* genes in photosynthesis, several of them are absent or pseudogenized in heterotrophic species of the genera Orobanche and the family Orchidaceae [21,22,23]. However, *ndh* genes were not found in some fully photosynthetic competent plants, such as the Gymnosperm *Pinus thunbergii* [24]. Except for Gnetales and several conifers, plastid DNA of Gymnosperms have *ndh* genes [25]. Thus, the plastid DNA of *Cycas taitungensis* [26] and of the Conifers *Cryptomeria japonica* (a Cupressaceae) [27] and *Thuja plicata* [28] contain the *ndh* genes. Frequently, *ndh* pseudogenes with large nucleotide deletions are found in the plastid DNA of plants lacking *ndh* genes, and comparison among phylogenetically close genera suggests that the functional genes were recently lost.

Among fully photosynthetic Angiosperms, species of several genera and families’ (e.g., *Erodium*, Ericaceae, *Najas*) *ndh* genes rarely correlate with taxonomic or evolutionary relationships [23] and, at least in the Orchids family, occurred after evolution into subfamilies [29,30,31,32,33]. Similarly, only two out of thirteen *Erodium* species (*E. texanum* and *E. carvifolium*) retain all eleven *ndh* genes intact in the chloroplast [29]. The *ndh* genes were reported to be absent in species of *Gentiana* sect. *Kudoa* [34], but they are present in species of *Gentiana* sect. *Cruciata* [35]. Extensive sequencing of plastid DNA within families and genera over the past few years frequently reports the absence of *ndh* genes in a few plants. As a recent example, of 25 complete DNA sequences from the genera *Aragoa*, *Littorella*, and *Plantago* of Plantaginaceae, only those of the aquatic genus *Littorella* lack *ndh* genes [36].

Independent losses of *ndh* genes have been found in families of the order Alismatales, where 10 of 94 plants tested lack *ndh* genes [37], in the family Tofieldiaceae, in the aquatic species *Najas flexilis* of the family Hydrocharitaceae [31], and specifically, in *Capparis spinosa* var. herbacea of the genus *Capparis* [38]. In contrast, through chloroplast DNA re-ordering, some Ericales duplicate six *ndh* genes (in the inverted repeat regions), while they lose one copy of the usually duplicated *ndhB* gene, and the *ndhF* remains alone in the small single-copy region [39,40] (compare with Figure 1). Not surprisingly, the loss of *ndh* genes and the inverted repeat region are found in some Cactaceae [41] and may be associated with crassulacean acid metabolism.

Representative phylogenetic trees for Ericaceae (Figure 3 of Reference [30]), *Erodium* (Figure 1 of Reference [29]), and Alismatales (Figure 3 of Reference [37]) show that losses of chloroplast DNA *ndh* genes from some Angiosperms appear to be recent (less than 50 Ma) and independent evolutionary events after most species’ diversification occurred. Possibly, most of the older losses of *ndh* genes produced plants unable to evolve and diversify. Plant species that have lost the *ndh* genes could be endpoints of the evolutionary tree and will become extinct. Arguably, many Angiosperms lost the *ndh* genes in mild environments, where the Ndh complex was dispensable [10,23,32,42,43,44,45]. They and their offspring were unable to survive and diversify in variable stress environments. Probably, the same considerations are also valid for Gymnosperms, where ongoing intensive sequencing programs for *Pinus* species, which probably diverge only less than 10 Mya [46,47], should clarify what species and how many contain *ndh* genes [48].

## 3. Functional Role of the Thylakoid Ndh Complex

Understanding the fate of the *ndh* genes during land plant evolution must be based on the reactions catalyzed by the Ndh complex in chloroplasts. These appear to be related to the cyclic photosynthetic electron transport and photophosphorylation. The Ndh complex is found in the stromal thylakoids [9,49,50] and catalyzes an oxidoreduction reaction whose electron acceptor is oxidized plastoquinone (PQ). There are some discrepancies about the identity of the electron donor. The similarity to respiratory complex I, as well as in vitro and zymogram assays, suggests that NADH is the electron donor [8,9,17,28,51,52,53], in accordance with the reaction:NADH + H^+^ + PQ → NAD^+^ + H_2_PQ

NADPH has no or negligible donor activity in assays with most plant preparations. Therefore, the Ndh complex would provide a pathway for PQ reduction not dependent on photosynthetic electron transport, that would initiate (Figure 2) a chlororespiratory electron transport chain [9,50,53]. Through dynamic oxidoreduction of plastoquinone, chlororespiration adjusts (poises) the redox levels of intermediaries [9,54,55,56] to optimize [57] cyclic electron transport (CET) following photoinhibition of photosystem II, under fluctuating light intensities or when temperature or humidity strongly affect CO_2_ availability or the rate of its reduction. CET complements linear electron transport (LET) (especially under different stress conditions and high CO_2_ concentrations) to polarize the thylakoid membrane, which is required to synthetize ATP and dissipate the excess of energy from excited chlorophylls (non-photosynthetic quenching of chlorophyll fluorescence, NPQ) via zeaxanthin (xanthophyll cycle).

At a sudden decrease in light intensity, the reductive CO_2_ cycle drains more electrons than PSII can supply, and the transporters would be over-oxidized if the Ndh complex does not supply extra electrons from the NADH formed, for example, by malate dehydrogenase. Conversely, at high light intensity and a low rate of CO_2_ fixation (due to low temperature or stomate closure under dry conditions), the electron transporters are reduced in excess, and electrons are drained by the Mehler reaction, producing anion superoxide (O_2_•^−^). The additional electrons are drained by the subsequent action of superoxide dismutase and plastoquinol peroxidase, which scavenge superoxide and hydrogen peroxide (H_2_O_2_) respectively, keeping the level of reactive oxygen species (ROS) under control.

Terminal oxidase [50,55] could be an additional electron-draining process of cyclic over-reduced electron transporters. The combined actions of the Ndh complex and oxidative reactions constitute the chlororespiratory electron transport chain that rapidly buffers the redox shifts of electron transporters while maintaining active CET. The balanced ratio of Ndh and oxidative reactions prevents the burst of ROS levels that can led to cell death [58,59,60,61,62,63,64,65].

Based on assays with *Arabidopsis* membrane preparations, Yamamoto et al. [66] reported evidence for reduced ferredoxin as an electron donor and postulated that the Ndh complex transports electrons from photosystem I to PQ, providing a route of CET additional to that of the commonly accepted model in which ferredoxin donates electrons to the intermediary electron pool, PQ/cyt.*b_6_f* [67]. Be that as it may, the involvement of the Ndh complex optimizing CET and the associated photophosphorylation is widely accepted.

## 4. Dispensing with the Role of the *ndh* Genes

The involvement of the Ndh complex optimizing photosynthesis would explain the absence of the plastid *ndh* genes in parasitic plants and some carnivorous plants [68,69] that rely on low or no photosynthetic activity. Logic suggests that unused genes would accumulate mutations and, eventually, be eliminated to economize plant chemical and energy consumption [10,23,32,42,43,44,45]. Among the mutations, T-to-C mutations are frequently corrected by C-to-U editing at the RNA level of the transcript [42] and, less frequently, by C-to-T reversion in DNA [41]. However, the pseudogenization and deletion stages of *ndh* genes should be affected by the effects of environmental changes on the evolution of other functional traits of plants. Thus, detailed analysis of gene loss in Orchid and Ericaceae species [21,30] revealed that *ndh* genes were among the first pseudogenized genes in the chloroplast during the evolutionary transition from phototrophic or mycoheterotrophic to wholly heterotrophic metabolism, raising the question of whether species without *ndh* genes would survive for a long time without the development of heterotrophic structural and functional adaptations.

Transgenic plants defective in *ndh* genes point to some clues based on the role of the Ndh complex in protection against different stresses. Increased expression of *ndh* genes under different environments provides additional lines of evidence. Thus, transgenic tobacco plants whose *ndh* genes have been inactivated show impaired photosynthetic activity [17,70], especially under fluctuating light intensities and high atmospheric CO_2_ concentrations [56].

Plants under stress demand extra ATP consumption. Consequently, to satisfy it, heat stress was reported to increase CET and photophosphorylation in grape leaves [71]. The plastid Ndh complex is an efficient proton pump that increases CET and associated phosphorylation [72]. More specifically, chlororespiration has been found to increase under stress [55,73,74,75,76] and protects against photo-damage of oxygen-evolving complex and PSII [77,78].

Many biochemical and functional assays using transgenic plants indicate that the Ndh complex improves photosynthesis efficiency, decreases entropy production [44,56], and protects the leaves against a variety of stresses. However, extensive research using plant species lacking *ndh* genes is required to confirm the selective advantages provided by the Ndh complex. In this line of research, Sun et al. [79] reported intense variability and loss of *ndh* genes in the critically endangered *Kingdonia uniflora*. Similarly, Folk et al. [80] found intense pseudogenization and loss of *ndh* genes in the semi-aquatic plant *Saniculiphyllum guangxiense*, which contrasts with the strong conservatism of the plastid genes in other Saxifragales. Plausibly, the precariousness of *Kingdonia uniflora* and *Saniculiphyllum guangxiense* could be related to the poor photooxidative protection due to the absence of *ndh* genes. Plants lacking *ndh* genes have more difficulty adapting to changing environments. Thus, while the non-invasive weed *Mikania cordata* lacks the *ndhF* gene, the invasive *Mikania micrantha* retains it [81] and could invade new environments.

Current explanations assume that *ndh* genes could be dispensed with in mild environments. However, their loss only slowly drove plants to the heterotrophic alternative or, eventually, to extinction when abiotic stress episodes affected terrestrial habitats [82]. Be that as it may, gene dispensation implies that plants without *ndh* have some evolutionary advantages over plants with *ndh* in mild environments. Metabolism economy may be one advantage, but not necessarily the only one.

Being involved in stress protection, comparative analyses of plastid genes in different plants frequently report positive selection of the *ndh* genes [83,84,85]. However, the subtle function of *ndh* genes makes it difficult to functionally and ecologically compare species that differ only by the presence of *ndh* genes. Moreover, in stress protection, several activities interact with that of the Ndh complex in different ways, which involve ROS signaling and transcriptional factors, and may result in either a protective response [86] or cell death [63,64,87]. The Ndh complex and *ndh* gene transcripts increase early during leaf senescence [9,59]. Accordingly, transgenic tobacco plants defective in the *ndhF* gene show delayed leaf senescence [62] (Figure 3), and chloroplast-related ROS activities are required for senescence and cell death in different plant systems [28,88,89,90,91,92].

Due to their involvement in ROS metabolism, *ndh* genes participate in complex crossroads regulating defense response, aging, and programmed cell death [63,64,65], three related cellular possibilities critical for plant survival and evolution.

*ndh* gene losses may have occurred frequently during the evolution of land plants, but the co-existence of *ndh*-less and *ndh*-containing plants of the same or newly diversified genera suggests: (1) that plants that lost *ndh* genes a long time ago (e.g., 10 or more Mya) became extinct, and (2) that the ancestors of extant *ndh*-less plants lost *ndh* genes recently. Therefore, *ndh* gene pseudogenizations are recent events in plant evolutionary trees that only include data from extant plants. An obvious conclusion would be that plant species lacking *ndh* genes are in danger of extinction.

## 5. Plant Death or Species Extinction, a *ndh* Dilemma?

Extinction of species lacking *ndh* genes requires recurrent periods of stress in most lands and over many generations. Their survival requires poorly investigated mechanisms of protection alternative to *ndh* genes, or a drastic decrease of ROS-generating metabolism, such as photosynthesis, as occurs in the heterotrophic metabolism of epiphytic and carnivorous plants.

Gnetales and some *Pinus*, which could have lost the *ndh* genes early in evolution, require further functional investigations and sequencing of the plastid DNA from many closely related species and subspecies.

Gnetales and some *Pinus* that lack *ndh* genes are surprisingly long-lived. The Gnetal *Welwitschia mirabilis* is a well-documented case of a plant lacking *ndh* genes [93]. It is the only species in the only genera of the Welwitschiaceae family of Gnetales, and estimates put its lifespan at up to 1000 years. Several *Pinus*, such as some individuals of *Pinus longaeva*, probably live for longer periods, and their needles remain alive and photosynthetically active for up to 30 years. As long-lived plants, they have surely suffered many periods of stress, but have survived without *ndh* genes. On the other hand, their longevity could be due in part to the absence of *ndh* genes, as found in transgenic tobaccos. To answer this question and, in general, the relationships among plant longevity, species survival, extinction, and diversification, requires detailed investigations on the distribution of *ndh* genes and pseudogenes in Gymnosperms and on the molecular mechanisms that protect long-lived species against stress.

In Angiosperms’ leaves, ROS not only destroy cell components, but are also transduction signals within the complex networks of molecules that modulate the response to various stresses and cell survival and death [94]. The destructive and signaling actions of ROS co-exist in responses to stress, aging, and senescence, and it is often difficult to distinguish between the two actions and among the three responses. Three major ROS, singlet oxygen (^1^O_2_), superoxide anion radical (O_2_•^−^), and hydrogen peroxide (H_2_O_2_), are effective wreckers of cellular components and are also cellular signals that affect gene expression and enzymatic activities. In general, ^1^O_2_ and O_2_•^−^ appear to promote cellular aging and programmed cell death (senescence), and H_2_O_2_ promotes the stress defense response [64,65,95,96]. In chloroplasts, ^1^O_2_ is generated by transfer of an electronic excitation from chlorophyll to O_2_. O_2_•^−^ is generated by transfer of an electron from reduced intermediaries to O_2_ or to ^1^O_2_. The Ndh complex, in concerted action with electron-draining reactions (chlororespiration, including superoxide dismutase, SOD), keeps the thylakoid membrane polarized, which allows dissipation as heat of excess energy from excited chlorophylls and decreases the formation of ^1^O_2_. SOD delays senescence by removing O_2_•^−^ and forming H_2_O_2_. Therefore, low levels of SOD are associated with senescence in plant [58,59,61,97,98,99,100] and animal [101] tissues. In contrast, mRNA translatable expression of *ndh* genes increases early in leaf senescence, and the Ndh complex increases during tissue senescence and fruit ripening [9,11,28,59,91,102,103]. Increased Ndh with lower SOD levels in senescence and aging, far from balancing the redox level of transporters, increases their reduced forms, thus hindering thylakoid polarization and dissipation of excess light energy, and worse, increasing ^1^O_2_ and O_2_•^−^ formation, as observed during barley flag leaf senescence [104]. Therefore, SOD and Ndh complex responses, as photosynthetic tissues age, emerge as key determinants of the leaf fate towards death (increased Ndh and decreased SOD) or survival.

Although the presence of *ndh* genes in nuclear DNA and their encoded polypeptides in chloroplasts cannot be excluded, Gnetales and some *Pinus* probably lack the Ndh complex. In addition, investigations on SOD and other activities that may be involved in protection of photosynthetic machinery in Gymnosperms are needed to understand their survival and longevity, often in extreme environments.

The molecular mechanisms of senescence have been investigated mainly in leaves of monocarpic Angiosperms and show great similarity with the programmed cell death (PCD) investigated in animals and plants [62]. When the tissue reaches an advanced, but poorly defined, stage of development, the death program triggers the expression of senescence-related genes involved in the ordered macromolecular breakdown, leading to cell death. In animal and non-photosynthetic plant tissues, mitochondria are critical in triggering PCD through cellular signals, including ROS, that re-program gene expression from live metabolic processes to cell death [105,106,107]. Often, the ROS damage makes it difficult to distinguish between ROS-mediated PCD and ROS-mediated cellular aging, in which the cell dies by accumulation of hazardous damages of cellular components. In photosynthetic tissues, chloroplasts replace mitochondria as the signal factory (including ROS) that triggers PCD and/or cellular aging [62]. Most responses to biotic and abiotic stresses are also mediated by ROS produced by mitochondria [108] and chloroplasts [12], adding further complexity to the time-course of death and defense responses in plants that are permanently exposed to variable environments. However, the molecular mechanisms of death and defense responses are reasonably well-known in Angiosperms, but are scarcely known in most Gymnosperms, where the absence of *ndh* genes in some of them is a factor to be considered.

The absence or low expression of *ndh* genes could explain why needles and the whole plant in several Gymnosperms do not appear to show PCD but show aging by accumulation of hazardous damages. Intriguingly, evidence links heritable deficiencies of complex I (the mitochondrial homologous of Ndh complex) to human longevity [109], in clear correspondence with the longevity observed in Gymnosperms lacking *ndh* genes. However, this does not explain why, without *ndh* genes, Gymnosperms can survive the stresses to which Angiosperms are supposedly protected by *ndh* genes. Further molecular and physiological investigations should shed light on the uncertainties surrounding the life extension of some plants, the dispensation of the *ndh* genes, and the extinction of most *ndh*-less plants.

## 6. Concluding Remarks and Future Perspectives

Recent analyses of the absence of plastid *ndh* genes in a number of plants and comparison with phylogenetic trees, as well as with functional investigations of the role of the *ndh* genes, allowed us to advance some conclusions and propose open questions about the functional and evolutionary consequences of the presence of the *ndh* genes.

The *ndh* genes allow Angiosperm species to survive in many stressful terrestrial environments and to maintain efficient photosynthesis. They provide little or no advantages in mild environments, where they consequently accumulate mutations and, eventually, become pseudogenes and are deleted from plastid DNA. Angiosperms that have lost plastid *ndh* genes survive in mild and moderate stress environments or adopt a heterotrophic or carnivorous metabolism that compensates for their low or absent photosynthetic efficiency. Variable environments along the scale time of biological evolution place Angiosperms without *ndh* at permanent risk of extinction, as may be occurring with the endangered *Kingdonia uniflora* [79]. Consequently, phylogenetic analyses indicate that Angiosperm species lacking plastid *ndh* genes lost them recently (typically less than 10 Mya). Most hypothetical species that lacked *ndh* genes more than 10 Mya became extinct, and the extant Angiosperms without *ndh* are probably evolutionary endpoints on phylogenetic trees.

In Gymnosperms, the still small number of available sequences makes a comprehensive phylogenetic analysis of *ndh* gene loss difficult. Although a low relevance of PCD is assumed for leaves and the whole plant in Gymnosperms, the long lifespan of many of them poses formidable problems of understanding from our knowledge of *ndh* genes and stress responses in Angiosperms. Therefore, in addition to advances in plastid and nuclear DNA sequencing, future research should reveal the mechanisms of the stress response in Gymnosperms, from the molecular and cellular levels, including ROS generation and scavenging, to membrane-protecting mechanisms. A satisfactory understanding of the dispensability of the *ndh* genes may only be achieved with a broad view that includes the peculiarities of Angiosperms and Gymnosperms in terms of photosynthetic metabolism, stress response, longevity, reproduction, diversification, and vulnerability to extinction, that could be affected by the loss of *ndh* genes.

## Figures and Tables

**Figure 1 ijms-22-12505-f001:**
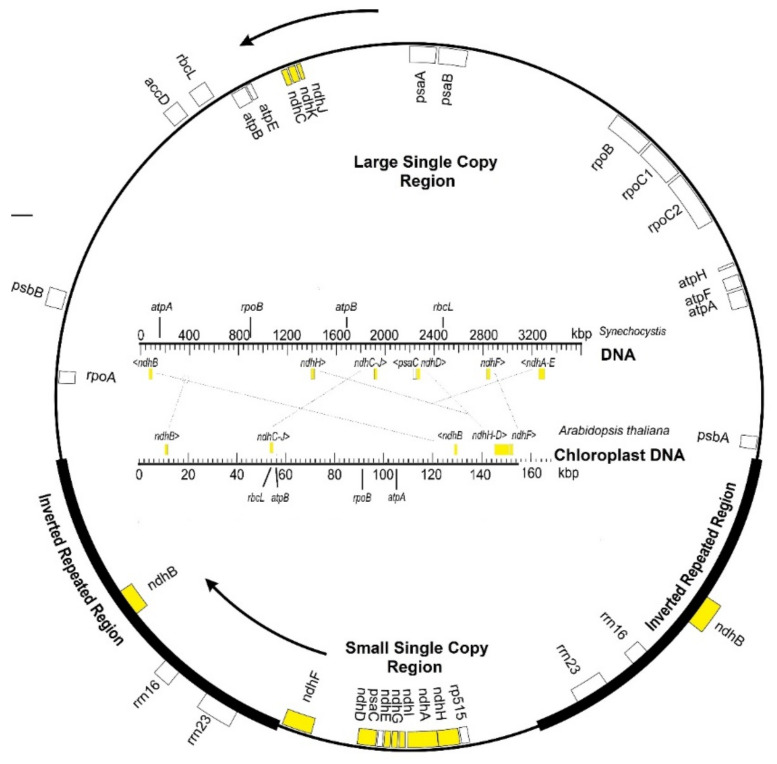
Map of *ndh* genes (yellow) in the circular plastid DNA typical of higher plants. Some other representative genes are also indicated. Those inside the circle are transcribed clockwise (inner arrow) and those on the complementary strand counterclockwise (outer arrow) and are depicted outside. The thick lines in the circle correspond to the inverted repeated regions. Inside the circle, the *ndh* gene map in *Arabidopsis* is compared with the homologous gene map of *Synechocystis* as a model cyanobacterium.

**Figure 2 ijms-22-12505-f002:**
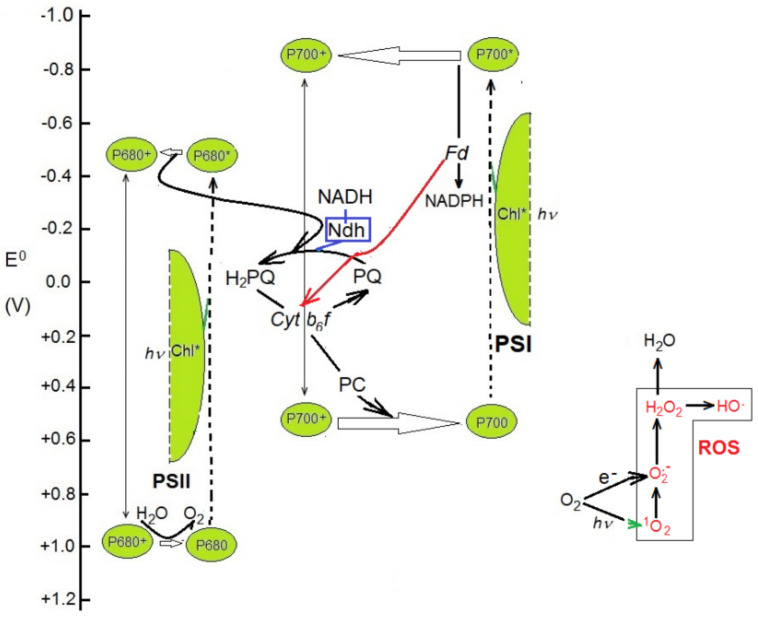
Connection of the Ndh complex with the photosynthetic electron transport. Electron transporters are displayed on the reference scale of the redox potential (E^0^). Arrows are marked in red for cyclic-specific electron transport (CET), in blue for electrons through the Ndh complex, and in green for electron excitation transfer. Box on the bottom right schematizes the main transformations of reactive oxygen species.

**Figure 3 ijms-22-12505-f003:**
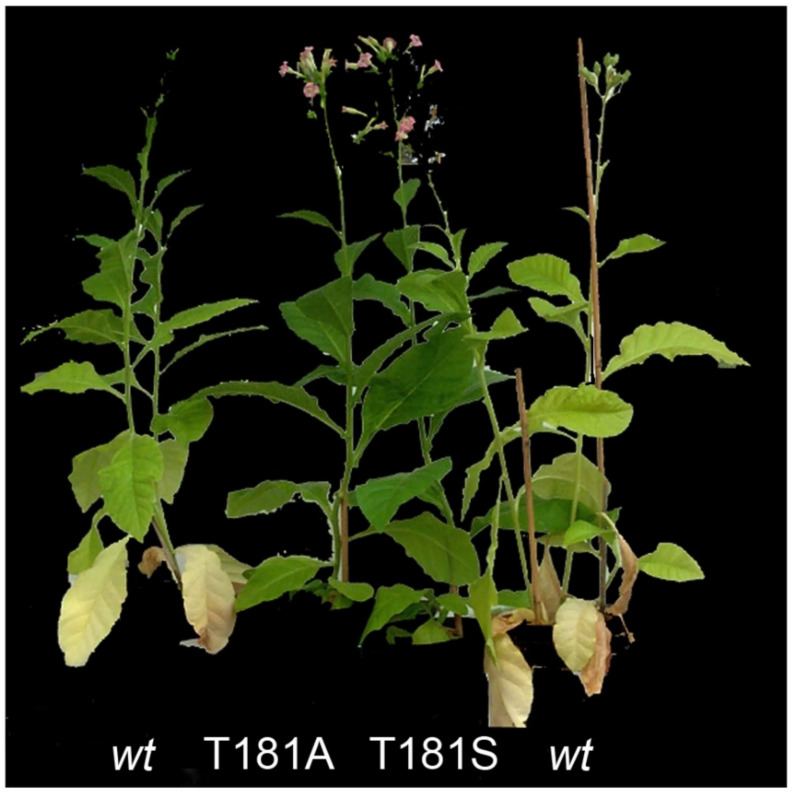
Delayed leaf senescence in T181A and T181S tobacco as compared with *wt* (Petit Havana), showing basal leaf senescence even before blooming. Tobacco T181A and T181S are point mutants obtained from wt tobacco in which the phosphorylable threonine at position 181 of the NDH-F subunit of the Ndh complex is changed to non-phosphorylable alanine and serine, respectively [17].

**Table 1 ijms-22-12505-t001:** Chloroplast *ndh* genes, encoded polypeptides, and homologous subunits in the respiratory complex I.

*Ndh* Gene	Encoded Polypeptide	Homologous Polypeptides in Respiratory Complex I (References)
*ndhA*	NDH-A	ND1/NuoH/FpoH, EchB, NQ08 [11,13]
*ndhB*	NDH-B	ND2/NuoN/FpoO, NQO14 [13,14,15]
*ndhC*	NDH-C	ND3/NuoA/FpoA, NQ07 [14,15,16]
*ndhD*	NDH-D	ND4/NuoM/FpoM, NQ013 [13]
*ndhE*	NDH-E	ND4L/NuoK/FpoK, NQ011 [13,15,16]
*ndhF*	NDH-F	ND5/NuoL/FpoL, NQ012 [13,17]
*ndhG*	NDH-G	ND6/NuoJ/FpoJ, NQ010 [13,18]
*ndhH*	NDH-H	49(IP)/NuoD/FpoD, EchE, NQ05 [13,19,20]
*ndhI*	NDH-I	TYKY/NuoI/FpoI, EchF, NQ09 [13,20]
*ndhJ*	NDH-J	30(IP)/NuoC/FpoC, EchD, NQ04 [13,20]
*ndhK*	NDH-K	PSTT/NuoB/FpoB, EchC, NQ06 [13,20]

## Data Availability

Not applicable.

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
