# Peer review of "On the Edge of Dispensability, the Chloroplast ndh Genes"

_ijms, 2021, doi:10.3390/ijms222212505_

Round 1
Reviewer 1 Report
The reviewed manuscript entitled “On the edge of dispensability, the chloroplast ndh genes” provides an interesting summary of studies on plastome ndh genes. The manuscript is generally well written and chosen topics are interesting. However, some issues should be resolved before final acceptance for publication in IJMS.
It would be interesting for readers to present information about ndf presence (or loss) in different plant families including their location on the plastome (LSC, SSC or IR). It could be done in table format or on a phylogenetic tree.
Lines 118-119: The distribution of ndh genes in the family Ericaceae, should be better discussed, since it is quite interesting, in some species like Vaccinium macrocarpon (Fajardo et al. 2012) or Chamaedaphne calyculata (Szczecińska et al. 2012), the SSC is formed by ndhF gene only, and rest of ndh genes which usually in SSC are move to IRs.
Author Response
The reviewed manuscript entitled “On the edge of dispensability, the chloroplast ndh genes” provides an interesting summary of studies on plastome ndh genes. The manuscript is generally well written and chosen topics are interesting. However, some issues should be resolved before final acceptance for publication in IJMS.
It would be interesting for readers to present information about ndf presence (or loss) in different plant families including their location on the plastome (LSC, SSC or IR). It could be done in table format or on a phylogenetic tree.
Replay.
I agree, graphic presentation about the presence of ndh genes among plants would be interesting. However, plants that lack ndh genes are a minority, even among those whose plastid DNA has been completely sequenced. Usually, they are highlighted within the family phylogenetic contexts that barely allow extension to higher order phylogenies that would show patched extensive tree. Therefore, at the present amount of data, I now inform to readers (lines 136-137 of the revised version) of excellent and easily available phylogenetic trees of some families showing that loses of ndh gene are independent evolutionary events after most specie diversifications occurred.
Lines 118-119: The distribution of ndh genes in the family Ericaceae, should be better discussed, since it is quite interesting, in some species like Vaccinium macrocarpon (Fajardo et al. 2012) or Chamaedaphne calyculata (Szczecińska et al. 2012), the SSC is formed by ndhF gene only, and rest of ndh genes which usually in SSC are move to IRs.
Replay.
I agree, plastid DNAs of Ericaceae deserve more attention. I thank your valuable information. In addition to phylogenetic tree indicated at line 136, please see new texts in lines 131-135, and lines 206-211 (that enlarged former text there), and the new references ([39,40]).
I thank you again for your valuable suggestions that, I think, have significantly improved the manuscript.
Reviewer 2 Report
The author may have misinterpreted the lifeform of some of the species discussed. This may be a consequence of a misinterpretation of the meaning of the term epiphyte, a plant that grows upon another but is not parasitic. Several of the species discussed are parasitic, hemiparasitic, mycoheterotrophic, carnivorous or aquatic plants. Full parasites do not photosynthesise, and hemiparasites may obtain a large proportion of their carbohydrates from their hosts. Similarly, mycoheterotrophic plants parasitise fungi to obtain carbohydrates. The loss of ndh genes in non-photosynthetic plants would indicate a loss of selective constraints on these genes. The lack of ndh genes in aquatic plants may indicate the absence of the absence of a selective constraint to related to drought and temperature stresses in their environments.
There is no discussion of the loss of ndh genes of photosynthetic plants within the Cactaceae or other members of the Caryophyllales other than mentions of carnivorous plants. Some authors have related this to the presence of crassulacean acid metabolism in some of these species. This is not discussed by the present author.
Rather than being indicative of evolutionary end points as suggested by the author, the loss of ndh genes in many species across a wide range of orders within vascular plants suggests that these genes are readily lost when their retention is not constrained by environmental pressure.
The manuscript is not suitable for publication in IJMS in its present form.
Author Response
The author may have misinterpreted the lifeform of some of the species discussed. This may be a consequence of a misinterpretation of the meaning of the term epiphyte, a plant that grows upon another but is not parasitic. Several of the species discussed are parasitic, hemiparasitic, mycoheterotrophic, carnivorous or aquatic plants. Full parasites do not photosynthesise, and hemiparasites may obtain a large proportion of their carbohydrates from their hosts. Similarly, mycoheterotrophic plants parasitise fungi to obtain carbohydrates. The loss of ndh genes in non-photosynthetic plants would indicate a loss of selective constraints on these genes. The lack of ndh genes in aquatic plants may indicate the absence of the absence of a selective constraint to related to drought and temperature stresses in their environments.
Replay.
Thank you very much for your expertise advice on the lifeform vocabulary. I hope to have corrected at several lines of the manuscript the previous unappropriated use of the term epiphyte. In the new text the term heterotrophic is widely used except when the referred publication specifically indicates, mycoheterotrophic, epiphytic, carnivorous, …
I agree, the loss of ndh genes in aquatic and in non-photosynthetic plants indicates the absence of selective constraints and, I think, all along the manuscript makes it clear. Certainly, drought and temperature stresses are not constraint in aquatic plants.
There is no discussion of the loss of ndh genes of photosynthetic plants within the Cactaceae or other members of the Caryophyllales other than mentions of carnivorous plants. Some authors have related this to the presence of crassulacean acid metabolism in some of these species. This is not discussed by the present author.
Replay.
That you for the call on Cactaceae. Unfortunately, the function of the Ndh complex, if present, in Cactaceae and its relations with the crassulacean acid metabolism are poorly known. Following your suggestion, I now refer ([41]) the absence of the ndh genes in the cactus Carnegiea gigantea and compares it with the extensive deletion of plastid DNA region in the distant Ericaceae (new lines 133-135). I hope to bring attention of more researchers in this interesting field.
Rather than being indicative of evolutionary end points as suggested by the author, the loss of ndh genes in many species across a wide range of orders within vascular plants suggests that these genes are readily lost when their retention is not constrained by environmental pressure.
Replay.
Of course. The ndh genes are readily lost when their retention is not constrained by environmental pressure. But: more, similarly, or less readily lost than other genes? And, probably more relevant, what possibilities of endurance and diversification have plants that lack ndh genes if environmental pressure came back. Their distribution in phylogenetic trees suggest low possibilities.
I thank you again for your suggestions and stimulating discussion.
Round 2
Reviewer 2 Report
The manuscript now contains a more comprehensive discussion. Some errors of expression require correction before publication. I have highlighted these in the attached document and provided explanations below. Where a word or letter is highlighted, but no explanation is given, just delete the unnecessary word or letter.

Author Response
Thank you very much for all careful suggestions and corrections to the manuscript that I have now incorporated. Please, a minor note: when compared with the pdf archive, I have assumed that your point for “L196” (add "s") is in fact for “L192”.